# Biochemical characterisation supports a central role for Oep80 from *Arabidopsis thaliana* in chloroplastic β-barrel protein assembly
Rhiannon J. Durant, R. Paul Jarvis & Jani R. Bolla ✉

The outer membranes of Gram-negative bacteria and endosymbiotic organelles mitochondria and chloroplasts contain β-barrel proteins essential for transporting metabolites, ions and proteins, as well as regulatory functions. While the assembly mechanisms of these proteins are well-studied in bacteria and mitochondria, they remain poorly understood in chloroplasts due to challenges in producing sufficient quantities of relevant plant proteins for structural and biochemical analyses. Here, we show that Oep80, the presumed core component of the chloroplastic β-barrel assembly machinery, can be overexpressed in bacteria and refolded in large quantities. While the soluble POTRA domain caused the heterologous protein to aggregate, we found that the isolated β-barrel membrane domain (Oep80$^\beta$) is very stable and refolds well. Using native mass spectrometry, we further show that purified Oep80$^\beta$ binds predicted β-signals found in several substrate β-barrel proteins in an anti-parallel manner. Overall, our findings support a central role for Oep80 in chloroplast β-barrel biogenesis. They provide evidence that the chloroplastic β-barrel assembly machinery uses a β-signal to recognise its substrates, suggesting mechanistic parallels with the BAM and SAM complexes of Gram-negative bacteria and mitochondria, respectively.

Mitochondria and chloroplasts both arose via endosymbiosis from Gram-negative bacteria[1]. Consequently, all three share common features, including the presence of an outer membrane (OM) containing β-barrel proteins. These β-barrel proteins perform essential roles such as the transport of small metabolites, ions and proteins across the membrane as well as regulatory and signalling functions[2]. Mutation, misfolding or misassembly of these β-barrel proteins is often associated with cell death, disease or developmental defects[3]. Any attempts to harness or control the biogenesis of the β-barrel proteins will require a comprehensive understanding of the associated molecular mechanisms. In the past 10–20 years, considerable insight has been gained into the pathways of intracellular transport and insertion of β-barrel membrane proteins in Gram-negative bacteria and mitochondria. Indeed, β-barrel assembly has even been identified as a novel antibiotic target[4,5]. However, several questions regarding the complete mechanism for folding and insertion of β-barrels into the OM still remain (Fig. 1)[2,6,7]. Of even greater importance, the mechanism of β-barrel assembly in chloroplasts remains mostly uncharacterised.

Chloroplasts are responsible for the bulk of terrestrial photosynthetic primary production, and their proper function relies on importing thousands of nucleus-encoded proteins via the TOC-TIC machinery (Translocons of the Outer and Inner Chloroplast envelope membranes)[8–10]. This machinery is further regulated by the chloroplast-associated protein degradation complex (CHLORAD)[11]. The core components of TOC and CHLORAD are β-barrel proteins, namely Toc75 (which co-assembles with Toc159) and SP2 (p39), respectively[12]. Additionally, the chloroplast outer envelope membrane (OEM) contains several other β-barrel proteins (e.g., Oep21, Oep24, Oep37, Tgd4) required for metabolite and solute transport[13–17] adding to around 16 β-barrel proteins identified in *A.thaliana*. How these β-barrel proteins reach the chloroplast and assemble in the OEM is largely unknown. Recent studies indicated that the translocation of β-barrel proteins across the OEM is TOC-dependent while membrane insertion requires the essential Omp85 family protein, Oep80 (also known as Toc75-V)[18]. Homozygous knockout of Oep80 is embryonic lethal, while RNAi-mediated knockdown reduces levels of β-barrel outer envelope proteins (OEPs)[19,20]. Although Oep80 appears to be a central component of the

Molecular Plant Biology, Department of Biology, University of Oxford, Oxford, UK. ✉e-mail: jani.bolla@biology.ox.ac.uk

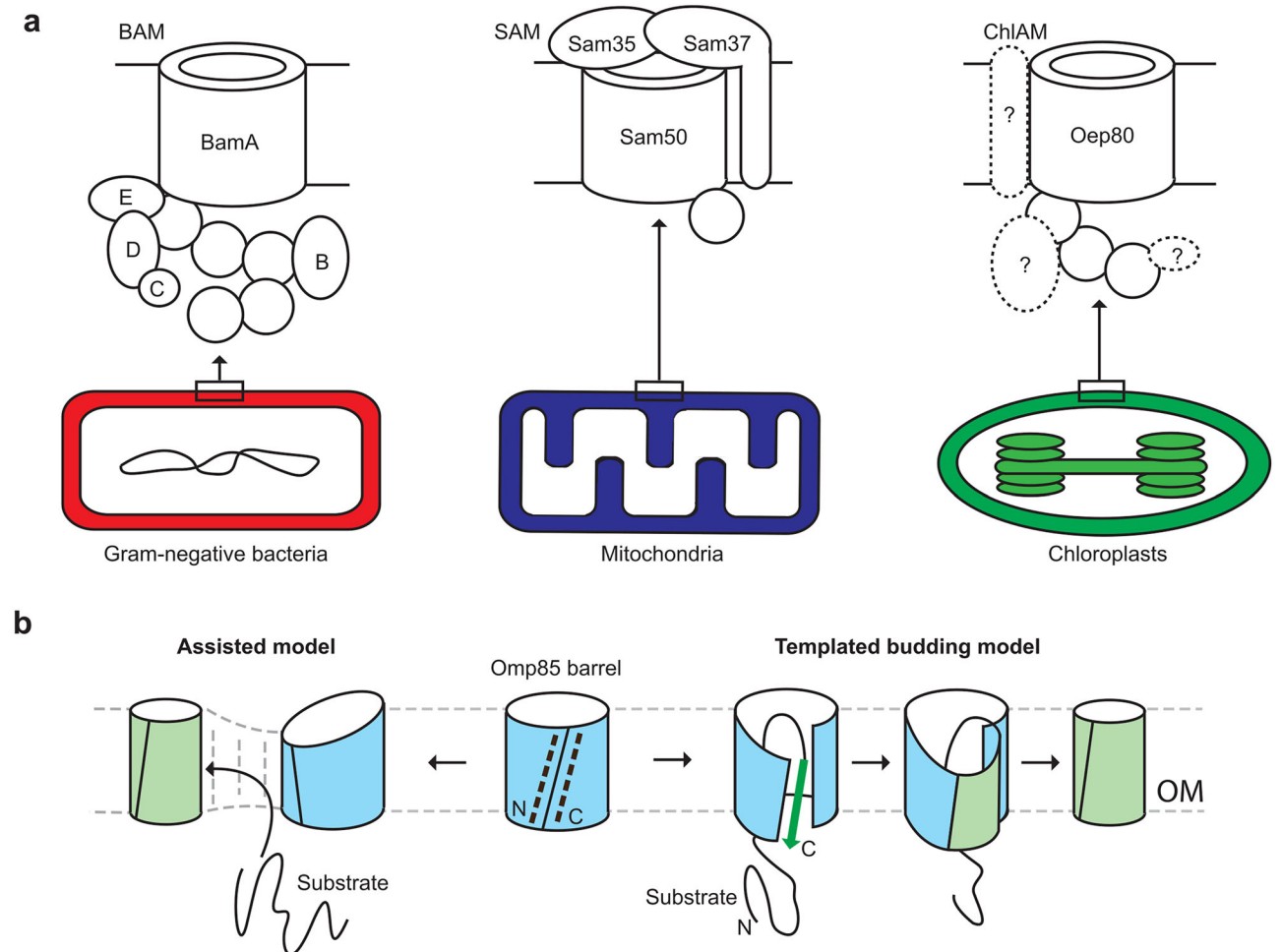

**Fig. 1 | The outer membranes of Gram-negative bacteria, mitochondria and chloroplasts contain β-barrel proteins, which require specialised machinery for assembly. a** Schematic diagrams of the β-barrel assembly complexes in each system. Question marks indicate assumed binding partners of Oep80 that have not yet been identified. **b** Two prevailing models for β-barrel assembly by Omp85 family proteins. N and C indicate N and C termini, respectively. Dotted black lines represent anti-parallel β-strands at the lateral gate. Green arrow represents the β-signal sequence of a substrate protein.

chloroplast β-barrel protein assembly machinery, other factors that assist in this β-barrel biogenesis process are assumed to exist and remain to be discovered. The membrane embedded protein Crumpled Leaf (CRL) has been shown to interact with Oep80, but whether it is directly involved with substrate folding is unclear[21].

In Gram-negative bacteria and mitochondria, β-barrel folding is predominantly performed by the BAM and SAM complexes, respectively (Fig. 1a). Both complexes contain a core Omp85 family protein, BamA or Sam50, composed of POTRA (polypeptide transport–associated) domains followed by a membrane-embedded β-barrel[3]. Omp85 barrels have a dynamic lateral gate, which enables recognition and binding of β-signal sequences to their first β-strand ($\beta_1$) (Fig. 1b)[6,22]. These sequences are found in the C-terminal β-strand of substrate barrels ($\beta_{-1}$). Sam50 recognises the consensus sequence PoXGXXφXφ, while BAM recognises XφXφXYXF (φ indicates hydrophobic, Po indicates polar)[23]. A recent study further demonstrated that internal sequences, termed internal β-signals, also contribute to substrate recognition by the BAM complex via interactions with BamD, suggesting a more complex, distributed recognition mechanism than previously envisaged[24]. Two major models exist for the mechanism of β-barrel assembly by Omp85 family proteins. In the assisted model, the shorter β-strands around the lateral gate causes thinning and instability in the adjacent membrane, which lowers the energy barrier for spontaneous folding and insertion of substrate barrel proteins (Fig. 1b)[2]. On the other hand, recent structural evidence supports a templated folding model, where barrel proteins fold strand by strand onto the $\beta_1$ strand of Sam50/BamA[25],

forming an extended hybrid β-sheet. This overcomes the energy barrier presented by the C-terminal strand's loss of hydrogen bonding with water. Once all strands of the new β-barrel have been incorporated, the first and last strands zipper together allowing the protein to bud off into the membrane (Fig. 1b)[26,27].

Import assays using chloroplastic β-barrel proteins Oep37 and SP2 showed their presence within Oep80-containing complexes, suggesting that Oep80 folds them into the OEM[18]. Additionally, mutating the antepenultimate hydrophobic residue and penultimate acidic residue in the $\beta_{-1}$ strand of Oep37 reduced the amount of Oep37 present in these Oep80-containing complexes, indicating these residues may be important for substrate recognition by Oep80. However, such a signal in other β-barrel proteins has not been identified. Identification of such a signal will help clarify whether all the chloroplastic β-barrel proteins biogenesis is mediated by Oep80; currently, there is no known alternative β-barrel assembly machinery in plastids. Furthermore, more evidence is needed to definitively show that Oep80 recognises and binds β-signals in substrate barrel proteins, and further characterisation of the β-signal consensus sequence is also required. Ultimately, mechanistic and structural analysis is needed to demonstrate that Oep80 forms the core of the chloroplastic β-barrel assembly machinery, and to determine whether it uses similar mechanisms to BAM and SAM to fold its substrates. In vitro structural and biophysical approaches are best suited for these analyses. Therefore, Oep80 must be overexpressed and purified from a heterologous expression system. So far, few chloroplast β-barrel proteins have been purified heterologously at large

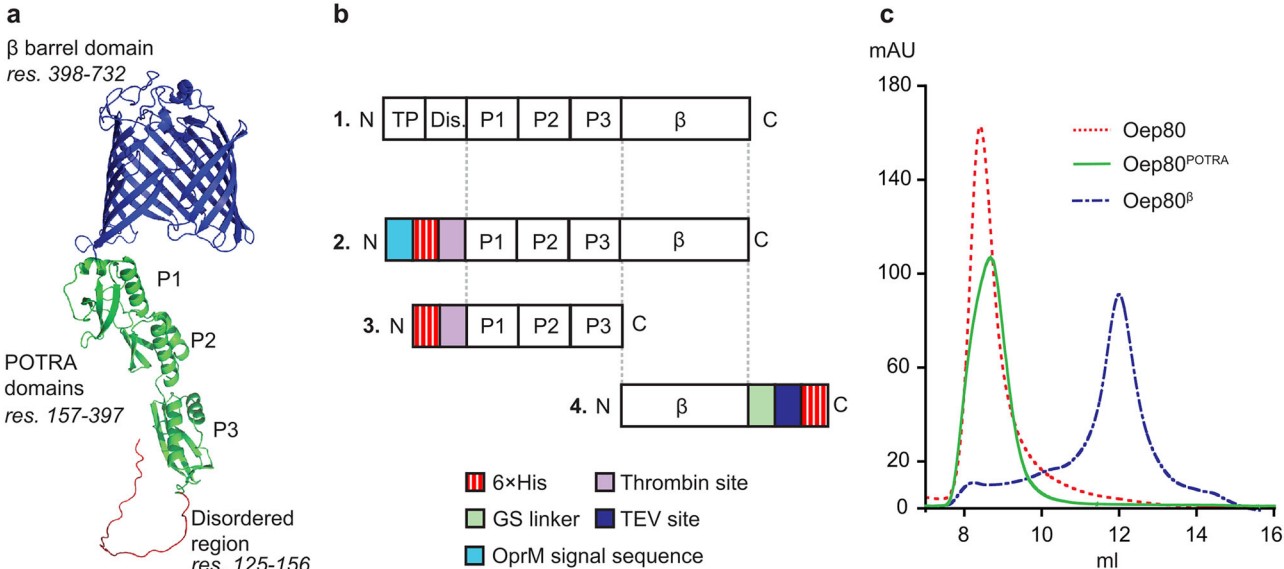

**Fig. 2 | Screening of various constructs indicated that the POTRA domain of Oep80 causes aggregation whereas the β-barrel domain alone can produce a monodisperse SEC peak. a** AlphaFold predicted structure of mature Oep80[35]. The cleavable transit peptide has been removed. **b** Domains of full-length AtOep80 (1.), plus constructs of Oep80 used to screen for refolding from *E. coli* inclusion bodies: 2 = 6×His-Oep80, 3 = 6×His-Oep80[POTRA], 4 = Oep80[β]-6×His. **c** Overlaid SEC traces of different Oep80 constructs. TP= chloroplast transit peptide, Dis.=disordered region, P = POTRA domain, β = β-barrel domain. N and C refer to the N- and C-termini, respectively.

scale. Most notably, the small channel protein Oep21 has been refolded from *E.coli* inclusion bodies for structural analysis by NMR[13]; and the barrel domains of Toc159, Toc75 and Toc132 have also been refolded and analysed by circular dichroism (CD)[28].

Here, we have screened various constructs and optimised the purification and refolding of milligram quantities of Oep80 from bacterial inclusion bodies. We observed that the soluble POTRA domain of Oep80 caused the full-length protein to aggregate. However, the refolded β-barrel domain (Oep80[β]) proved to be stable. Further analysis by native mass spectrometry (MS) and circular dichroism indicated that the protein is folded correctly, enabling us to test for β-signal interactions. The data show that the chloroplastic β-barrel assembly machinery very likely uses a β-signal sequence mechanism to recognise its substrates, revealing mechanistic similarity with its mitochondrial and bacterial counterparts, SAM and BAM.

## Results

### Heterologous expression of Oep80 in *E. coli* results in inclusion bodies

To produce Oep80 for in vitro studies, we generated a range of Oep80 constructs for heterologous expression in *E. coli* (Supplementary Table 1). These include affinity tags and fusion tags both at the N-terminus and C-terminus of the mature Oep80 (β-domain and POTRA domains) without the N-terminal transit peptide and the disordered domain (Fig. 2a). Additionally, we also made a construct with the OprM signal sequence at the N-terminus to target the Oep80 to *E. coli* outer membrane[29]. Next, we expressed all these constructs in several different cell lines including Arctic Express™, BL21(DE3) and BL21(DE3)PLysS. To our surprise, all these constructs led to expression in inclusion bodies only, even of the soluble POTRA domain (Supplementary Table 1). We then decided to establish a refolding process. For this, we selected the conditions and constructs that produced the greatest yield of inclusion bodies (Fig. 2b, Supplementary Fig. 1).

### Oep80 POTRA domain is a barrier to refolding

Prior to optimising refolding screens, we first solubilised the inclusion bodies in urea for all constructs and purified via affinity chromatography in urea. We then tested various refolding processes, including drop-wise

dilution, dialysis and on-column refolding with different detergents, salt concentrations, pH values, and additives including L-arginine. We noticed that drop-wise dilution worked well for most cases. Most of these tests resulted in no clear precipitation, indicating that milligram quantities of proteins can be refolded. However, when passed through size exclusion chromatography (SEC) the constructs including the β-domain and POTRA domains eluted at the void volume, suggesting misfolding and aggregation (Fig. 2c). We then sought to determine whether the POTRA or β-barrel domain was causing the protein to aggregate. For this, we attempted to refold a construct comprising only the soluble POTRA domains of Oep80 (Oep80[POTRA]). To our surprise, this protein eluted at the void volume in SEC (Fig. 2c), which contrasted with previous observations on the homologous Toc75 POTRA domains[30]. Varying refolding conditions and methods did not improve the resulting SEC trace for Oep80[POTRA].

In contrast, the constructs containing the β-barrel domain only (Oep80[β]) refolded well and showed a monodisperse peak beyond the void volume after SEC (Fig. 2c), indicating the POTRA domain was responsible for protein aggregation in full-length constructs. We next focused on further optimising the purification of Oep80[β] to enable in vitro biochemical characterisation.

### Oep80[β] refolds well in LDAO detergent

In order to produce Oep80[β] in milligram quantities, we used our optimised drop-wise dilution method from above and performed a refolding screen of Oep80[β] with various detergents including Lauryldimethylamine oxide (LDAO), n-dodecyl β-D-maltoside (DDM) and n-octyl-β-D-glucoside (OG) and subjected the refolded protein to SEC (Fig. 3a). The monodisperse peak is also observed in other detergents; however, the best conditions that yielded good amounts of refolded protein were 200 mM L-arginine, 300 mM NaCl and pH 8 buffer with LDAO as the optimal detergent. The presence of L-arginine considerably increased the maximum concentration of the urea-unfolded sample that could be refolded by drop-wise dilution without visible precipitation.

We also observed that Oep80[β] was sensitive to concentration using centrifugal ultrafiltration concentrators. Consistent with this observation, more extensive concentration by ultrafiltration prior to SEC led to a much larger proportion of the protein eluting at the void volume (Supplementary Fig. 2). To eliminate the need for ultrafiltration, cation exchange columns

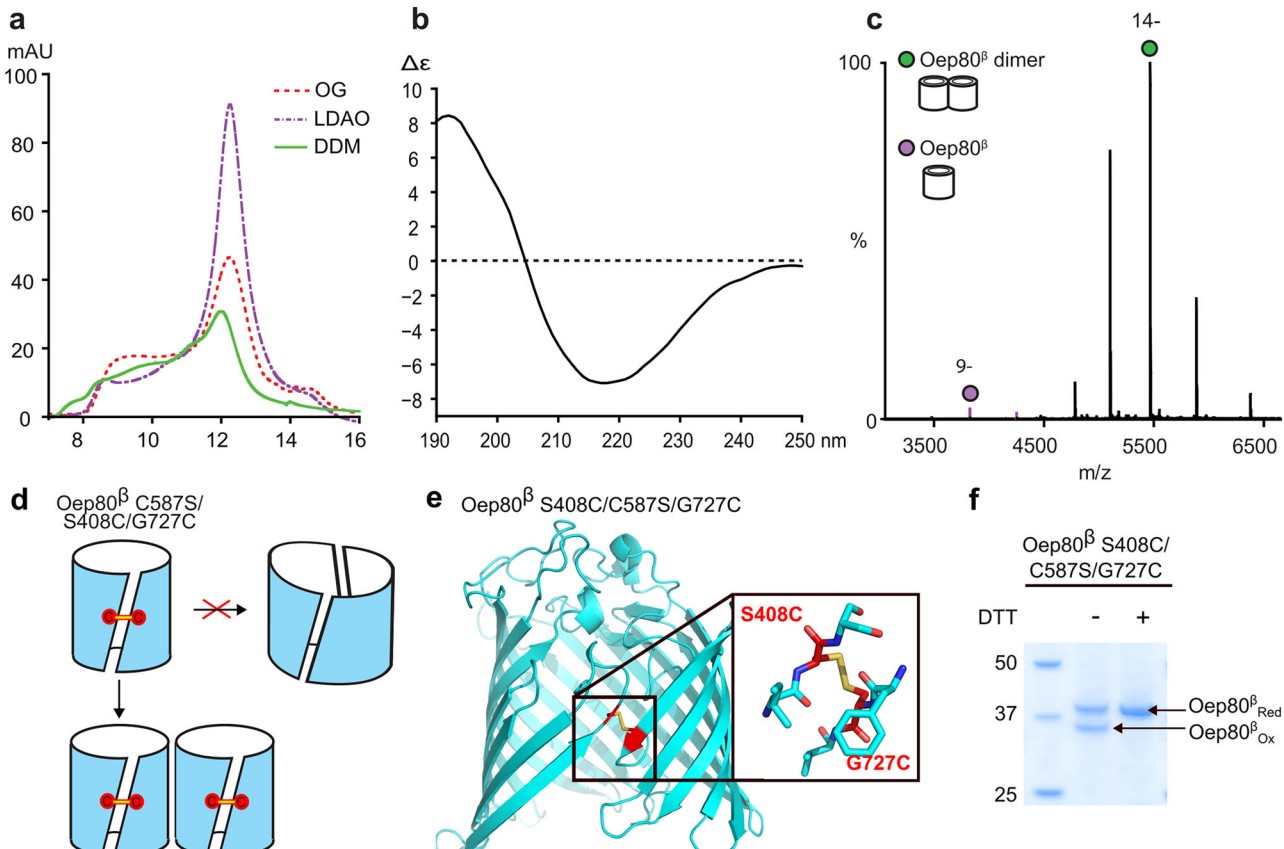

**Fig. 3 | Refolded Oep80[β] behaves as a folded β-barrel protein and exists as a non-hybrid barrel dimer in solution. a** Overlaid traces from SEC performed on Oep80[β] refolded using three different detergents. **b** CD spectrum performed on Oep80[β] in LDAO. **c** Native mass spectrum of Oep80[β]-6×His. For predicted and observed masses, see Supplementary Table 2. **d** Schematic to show a cross-gate disulfide should prevent hybrid barrel formation. Red circles represent introduced cysteine residues and yellow represents disulfide crosslinking. **e** AlphaFold 3 model of Oep80[β] C587S/S408C/G727C, showing a lateral gate-spanning disulfide bond in yellow. Introduced β[1] and β[-1] cysteines coloured in red. pTM = 0.84. **f** SDS-PAGE on Oep80[β] C587S/S408C/G727C post SEC purification and after incubation at 4 °C with 0.1 mM CuSO[4] for 3 h. +DTT indicates the sample was boiled with 50 mM DTT in loading dye.

were used to concentrate the protein to a smaller volume prior to SEC. We also noted that the Oep80[β] protein when concentrated in DDM, began to visibly come out of solution, even though DDM is a mild detergent and has been used for structural studies extensively.

### Purified Oep80[β] shows correct folding and exists as a dimer

To confirm that the purified Oep80[β] protein is folded properly, we subjected the sample from the SEC peak fraction to circular dichroism (CD) spectroscopy. The spectrum showed a minimum at 218 nm, which is as expected for β-barrel proteins (Fig. 3b, Supplementary Fig. 3)[31]. Moreover, thermal scanning CD at 218 nm showed a sharp transition, implying that the protein was initially folded, with a midpoint indicating a melting temperature of around 48 °C (Supplementary Fig. 4). We also used native MS to further confirm the extent of folding. After buffer-exchange of the sample into 200 mM ammonium acetate pH 8.0 containing 0.5% C8E4, the resultant mass spectrum showed the presence of two distinct charge state series with measured masses corresponding to monomeric and dimeric Oep80[β] (Fig. 3c, Supplementary Table 2). The tight distributions of these two charge state series is indicative of the protein being in a properly folded state. Our data also suggest that the protein is mostly dimeric in solution with <5% of monomers.

To confirm that there is no influence of the 6×His tag on dimerisation, we similarly analysed an N-terminally-tagged 6×His-Oep80[β] construct. This protein also showed dimerisation, both before and after thrombin cleavage to remove the 6×His tag (Supplementary Figs. 5 and 6). We then incubated tagged and untagged (post thrombin cleavage) Oep80[β] together and applied the mixture to Ni-NTA beads; after washing, the untagged protein co-eluted with the 6×His-tagged version under high imidazole but

eluted in the flow-through only in a negative control experiment lacking the tagged protein (Supplementary Fig. 7). This supported the view that dimerisation is an inherent property of the Oep80 β-barrel domain.

### Oep80[β] dimer is not a hybrid-barrel

To confirm the barrel closure and to determine whether the observed dimer forms a hybrid barrel or two independent β-barrels, we generated a mutant of Oep80[β] that has native cysteine C587 removed and 2 new cysteines introduced in the β[1] and β[−1] strands (C587S/S408C/G727C) (Fig. 3d). Mutation of BamA residues G433 and N805 to cysteine had previously been shown to form a gate-bridging disulfide[32]. Comparison of Oep80's Alpha-Fold model with BamA's known structure suggested S408 and G727, respectively, as the corresponding residues in Oep80. Strikingly, AlphaFold 3 spontaneously predicted a gate-bridging disulfide bond for the C587S/S408C/G727C mutant (Fig. 3e). After refolding and purification of this mutant, we then monitored disulfide bond formation on SDS-PAGE under reducing and non-reducing conditions[33]. A faster-migrating band was observed in the absence of DTT, consistent with the formation of an intramolecular disulfide bond between the added cysteines (Fig. 3f). This band disappeared upon DTT treatment. This confirmed that the cysteines are in close proximity in the folded state and form intra-strand disulfide bonds, which likely span the lateral gate. We also attempted to track folding kinetics using disulfide crosslinking but found that the need for reducing conditions during refolding complicated time-resolved quantification. Nevertheless, crosslinking becomes detectable within 10 min, suggesting rapid folding similar to that observed with dilution refolding of bacterial barrel proteins[34].

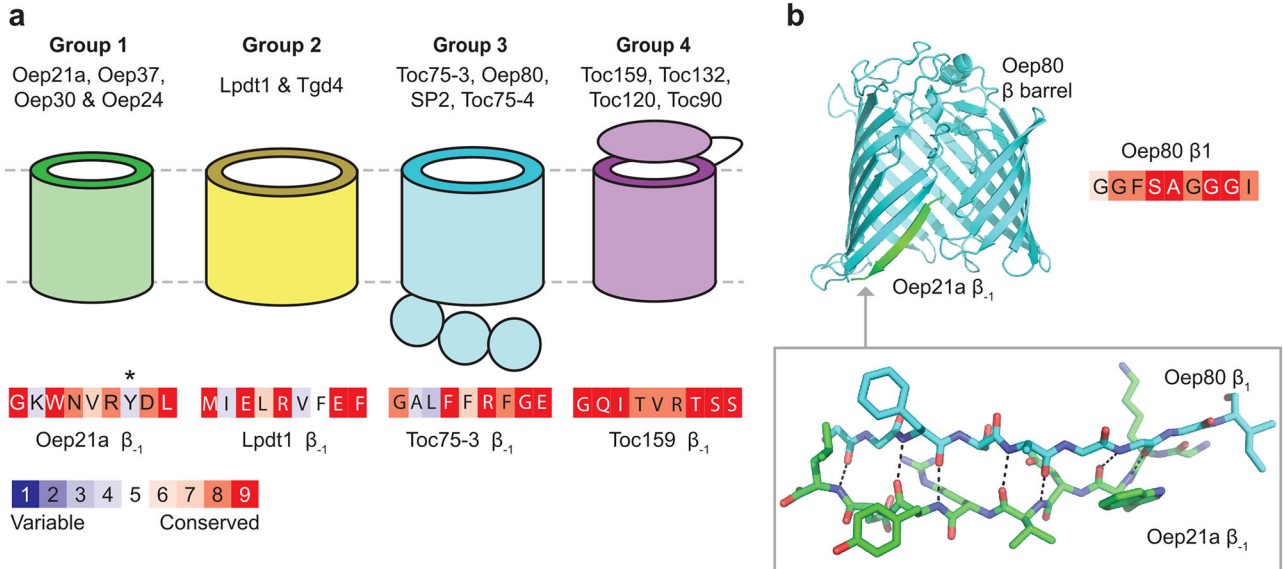

**Fig. 4 | Identification and bioinformatic analysis of putative β-signals in chloroplast β-barrel proteins. a** Outer envelope β-barrel proteins were assigned to four functional groups. Conservation analysis using ConSurf was performed on the predicted β-signal sequences, and a representative example is shown for each group. **b** AlphaFold 3 model of Oep80 interacting with the Oep21a $\beta_{-1}$ peptide (residues 159–167). IPTM = 0.6. Conservation analysis of the $\beta_1$ strand of Oep80 is also shown. Black dotted lines in the inset indicate backbone N-to-O distances of 3 Å or less, i.e., likely hydrogen bonding interactions. *Although conservation score is low for this residue, all 150 sampled orthologs have an aromatic side chain at this position (either Y or F).

In a hybrid-barrel configuration, crosslinking across the gate would block $\beta_1$ strand availability and thus be expected to interfere with inter-subunit strand exchange. If the observed dimer resulted from a hybrid barrel, we would expect $Cu^{2+}$-mediated disulfide formation to disrupt this interaction and shift the equilibrium toward the monomeric species (Fig. 3d). However, native MS analysis of Oep80$^\beta$ C587S/S408C/G727C both before (Supplementary Fig. 8a) and after (Supplementary Fig. 8b) $Cu^{2+}$ incubation showed no significant presence of a monomeric form of the protein. This suggests that the observed dimer is not a large hybrid-barrel but instead two independent barrels that are associated in solution.

### Identified OEP β-signal sequences show conservation and are predicted to interact with the $\beta_1$ strand of Oep80

Both BamA and Sam50, the core components of the BAM and SAM complexes, respectively, are known to use β-signal sequences to recognise their substrates. We asked if Oep80 does the same in chloroplasts. For this, we turned our attention to AlphaFold 3[35]. Firstly, we performed positive control experiments using BamA and Sam50 to test if AlphaFold 3 could predict binding of known β-signal peptides at the expected location. As expected, the models showed antiparallel β-sheet formation between the β-signal peptides and the $\beta_1$ strand of the barrel domain in both cases (Supplementary Fig. 9), matching what has been observed by structural determination and crosslinking studies[22,36].

Next, we predicted the structures of all known *A. thaliana* chloroplast outer envelope β-barrel proteins using AlphaFold 3 and split them into four groups by function (Groups 1–4). The nine residues making up the final β-strand of each barrel were then defined as the $\beta_{-1}$ strand and taken to be the presumed β-signal in each case (Supplementary Fig. 10 and Supplementary Table 3). In addition, ConSurf analysis was performed on these putative signals to provide an estimation of conservation and functional importance (Fig. 4a, Supplementary Fig. 11)[37]. Group 1 proteins are solute transporters, and previous work has shown they all share antepenultimate hydrophobic and penultimate acidic residues in the putative signal[18]. Group 2 proteins are lipid transporters of the LptD family, and their putative β-signals possess a similar hydrophobic/acidic motif[38,39]. Group 3 proteins are the Omp85 family homologues in Arabidopsis[40]. Finally, Group 4 proteins are the Toc159

family proteins which have N-terminal cytosolic domains and co-assemble with Toc75 within the TOC complex[12,41].

Next, we modelled the $\beta_{-1}$ strand sequences of *At*Oep21a and *At*Oep37 with *At*Oep80. These were chosen as previous work has suggested their *P.sativum* homologues interact with Oep80 and their $\beta_{-1}$ sequences were almost identical in *P.sativum* and *A.thaliana*[18]. In both cases, the peptide interacted via antiparallel β-sheet formation with the $\beta_1$ strand of the Oep80 barrel, paralleling the Sam50 and BamA control results (Fig. 4 and Supplementary Fig. 12). Of note, the iPTM (interface predicted template modelling) scores for Omp85-family protein interactions with cognate β-signals are relatively low and are in the order of 0.60–0.80 even for the controls.

Lastly, we selected a representative sequence from each of the four groups for further in vitro analysis (Fig. 4a).

### Purified Oep80$^\beta$ binds predicted β-signals of substrate OEPs

Our analysis above revealed variations in the putative β-signal sequences in different substrates. To test whether Oep80$^\beta$ can recognise these sequences, we used native MS. We selected representative members of each group—namely Oep21a and Oep37 for Group 1, LptD1 for Group 2, Toc75 for Group 3, and Toc159 for Group 4—and synthesised the respective $\beta_{-1}$ peptides. We combined 100 μM of each synthesised peptide with 7 μM of the purified Oep80$^\beta$ protein in C8E4 buffer and acquired native MS data (Fig. 5a, Supplementary Fig. 13, Supplementary Table 4). In all cases, in addition to the main charge state series, we observed adduct peaks corresponding to a mass consistent with the peptide binding to Oep80$^\beta$. Interestingly, the peptides of Oep37 and Oep21a showed much higher binding than those of Lptd1, Toc75-3 and Toc159 (Fig. 5a). Because the peptides differ in net charge and hydrophobicity, it is likely that they dissolve and ionise differently which could affect their apparent binding affinity in native MS. However, in all cases, peaks corresponding to the β-signal peptide alone were observed in the data, indicating that the peptides were present in solution and could be ionised. Additionally, analysis of the hydrophobicity and charge of each peptide at pH 8 did not reveal any pattern correlating with the data (Supplementary Table 5); for example, the hydropathy and charge of Oep21a $\beta_{-1}$ and Lptd1 $\beta_{-1}$ are almost identical, and yet Oep21a interacts more strongly with Oep80$^\beta$.

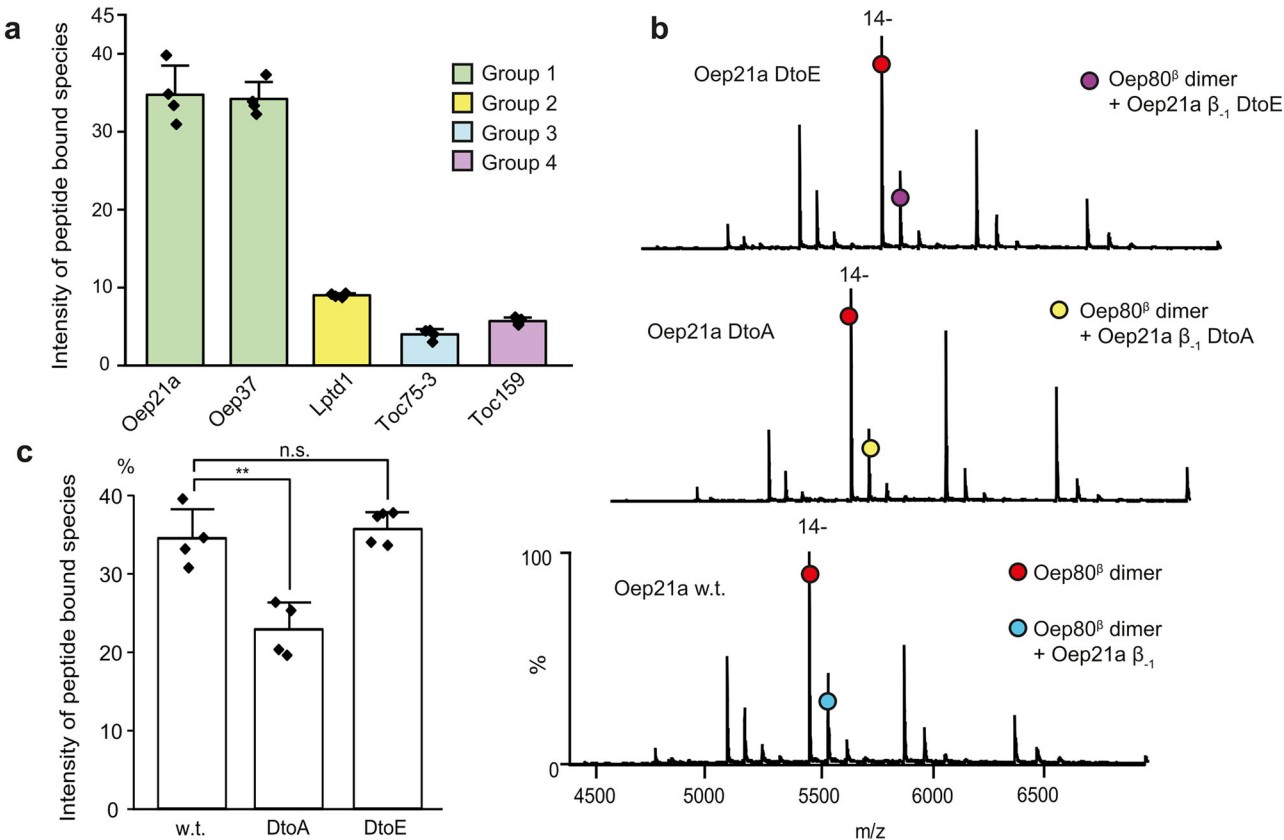

**Fig. 5 | Oep80 interacts with β₋₁ peptides of known substrate proteins with a penultimate acidic residue being important for the interaction. a** Average intensity of peptide-bound Oep80$^β$ relative to unbound Oep80$^β$ species for representative β-signal peptides from all four OEP groups, as measured by native MS. $n = 4$. **b** Native mass spectra of Oep80$^β$ plus Oep21a β₋₁ wild-type (w.t.) peptide (GKWNVRYDL), Oep21a β₋₁ D-to-A peptide (GKWNVRYAL), or Oep21a β₋₁ D-to-E peptide (GKWNVRYEL). See Supplementary Table 4 for predicted and observed masses. **c** Intensity of peptide-bound Oep80$^β$ relative to unbound Oep80$^β$ for Oep21a β₋₁ wild-type peptide and the two mutants. Significance was assessed using a two-tailed unpaired Student's $t$-test; n.s. = mean differences not significant, ** = means significantly different with $p < 0.01$ ($n = 4$). Error bars represent standard deviation.

Having found that Oep21a β₋₁ interacts strongly with Oep80$^β$, we used this peptide to further validate our conclusion that the Oep80$^β$ dimer does not form a hybrid barrel. We generated an Oep80$^β$ mutant with the Oep21a β₋₁ peptide sequence covalently attached via a flexible GS linker. This mutant also had the Oep80$^β$'s one native cysteine residue mutated to serine. In this construct, the covalently attached β₋₁ peptide is expected to form an anti-parallel β-sheet with the β₁ strand, mimicking the interaction observed in our peptide-binding native MS experiments. If the Oep80 dimer were a hybrid barrel formed by inter-subunit strand exchange, this steric occlusion should reduce dimerisation. Native MS revealed that this fusion protein also exists as a dimer (Supplementary Fig. 14). This reinforces the conclusion that the Oep80 dimer comprises two independently folded β-barrels that associate non-covalently, rather than a single hybrid structure formed by strand-swapping.

**Mutation of the predicted β-signal peptide sequence reduces interaction with Oep80$^β$**

Previous work has suggested that the penultimate acidic and antepenultimate hydrophobic residues of Group 1 and 2 outer envelope proteins are important for interaction with Oep80[18]. In particular, mutation of these residues in Oep37 was shown to reduce incorporation of the mutant protein into the outer envelope membrane. To probe whether these differences are related to the binding of these residues to Oep80, we generated mutants of the Oep37 β-signal sequence, one with a D-to-A mutation and one with a W-to-E mutation. We mixed these two mutants with Oep80$^β$ in separate experiments and acquired native MS data. In both cases, the resultant spectra indicated reduced peptide interaction, though the reduction observed was stronger when the acidic residue was mutated (Supplementary Fig. 15).

To confirm that this specificity was not unique to Oep37, we also mutated the penultimate residue of the Oep21a β₋₁ peptide, which is also D. Subsequent native MS analysis of this D-to-A mutant revealed a significant reduction in its interaction with Oep80$^β$. We then sought to confirm that the properties of the residue were important, rather than its exact identity, by analysing a D-to-E mutation. This mutation showed no significant difference from the wild-type sequence (Fig. 5b, c), indicating that negative charge at this position is important. These results collectively confirm the importance of charged and hydrophobic residues (at −2 and −3 positions, respectively) in the β-signal sequences.

**Oep21a β-signal peptide appears to bind in an anti-parallel fashion to Oep80's β₁ strand**

To directly test the location and orientation of β-signal peptide binding to Oep80$^β$, we generated a cysteine mutant of Oep80$^β$ (C587S/S408C), placing the cysteine at the C-terminal end of the β₁ strand. We then synthesised two versions of the Oep21a β₋₁ signal peptide: one with a C-terminal cysteine (CterC; GKWNVRYLC) and one with K160 toward the N-terminus mutated to cysteine (KtoC; GCWNVRYL). If the peptide binds to the β₁ strand in an anti-parallel orientation, as predicted by AlphaFold 3, only the KtoC peptide should be able to form a disulfide bond with the S408C site (Fig. 6a). This prediction was further supported by AlphaFold 3 structural models of the mutant Oep80$^β$ in complex with KtoC (Fig. 6b), which show a plausible lateral gate interaction and a predicted disulfide bridge between the introduced cysteines.

Consistent with this model, incubation of Oep80$^β$ C587S/S408C with each peptide in the presence of CuSO₄ to promote oxidation led to disulfide crosslink formation with the KtoC peptide but not with CterC (Fig. 6c). A

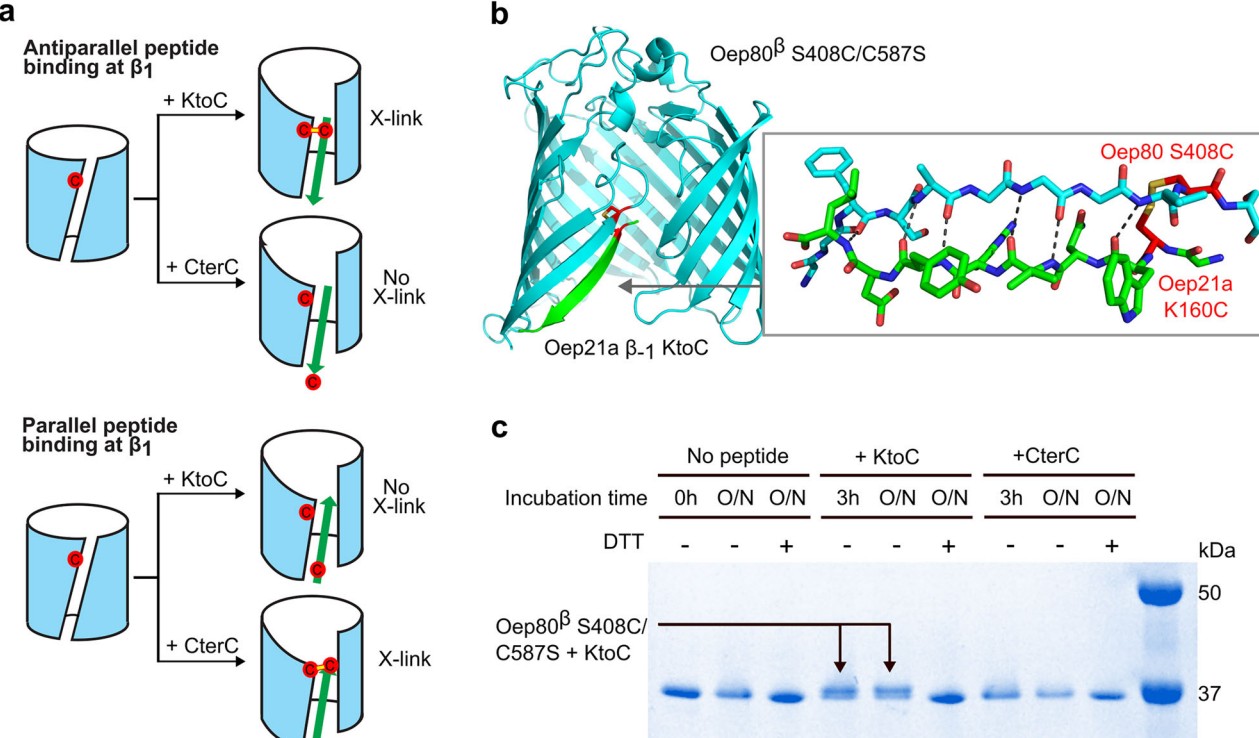

**Fig. 6 | Oep80$^\beta$ C587S/S408C crosslinks efficiently with Oep21a $\beta_{-1}$ KtoC but not Oep21a $\beta_{-1}$ CterC, indicating anti-parallel binding to Oep80's $\beta_1$ strand.**
**a** Schematic showing the experimental strategy. Oep80$^\beta$ C587S/S408C combined with mutant Oep21a $\beta_{-1}$ peptides containing cysteines on either end enables differentiation between parallel and antiparallel signal peptide binding. Green arrow represents peptide; red circles represent introduced cysteine residues and the yellow lines represent disulfide bonds. **b** AlphaFold 3 model of Oep80$^\beta$ C587S/S408C plus

Oep21a $\beta_{-1}$ KtoC. The inset shows the interaction interface in detail. Peptide backbone in green, Oep80 barrel backbone in cyan. Mutant residues coloured red. Dotted lines in the inset indicate likely hydrogen bonding interactions. **c** Oep80$^\beta$ S048C/C587S alone or plus either peptide was incubated at 4 °C with 0.1 mM CuSO$_4$. Samples were taken after 3 h (3 h) or overnight (O/N). DTT (+) indicates the sample was boiled with 50 mM DTT in loading dye.

higher molecular weight band was observed by SDS-PAGE only in the KtoC condition, and this band was reduced by DTT treatment, confirming disulfide bond formation.

To ensure the absence of crosslinking with CterC was not due to reduced binding affinity, we performed native MS analysis using both peptides. We observed the charge state series corresponding to both KtoC and CterC binding with similar intensity, indicating comparable non-covalent binding (Supplementary Fig. 16, Supplementary Table 6). This supports the interpretation that both peptides bind to the same site, but only KtoC positions its cysteine close enough to form a disulfide with Oep80 $\beta_1$-S408C.

## Discussion

In this study, we optimised the purification of full-length and various truncated forms of Oep80, a protein that has been implicated in the chloroplast β-barrel assembly machinery. Despite several attempts, we were unable to express Oep80 in its natively folded form. Previous work has shown that the OprM signal sequence should be most effective for directing barrel proteins to the bacterial outer membrane, where they may fold natively[29], but this did not result in folded Oep80 protein. It is possible that the protein is unable to fold into the OM upon arrival due to the latter's dissimilarity with the chloroplast OEM. The protein is also very likely to require assistance with folding, and the bacterial β-barrel assembly machinery may be unable to recognise its β-signal sequence. Therefore, expression into inclusion bodies and refolding were optimised. The POTRA domain of Oep80 appears to be prone to misfolding or aggregation, as we were unable to find refolding methods or buffer conditions that would result in non-void SEC peaks for this domain alone or the full-length protein. It is possible that the unique environment of the chloroplast inter-membrane

space, or assistance from specific chaperones, is required for correct folding of this domain.

In any case, it was possible to refold the isolated β-barrel domain of Oep80, Oep80$^\beta$. Analysis of refolded Oep80$^\beta$ by native MS, SEC and CD spectroscopy indicated that it is correctly folded, and that it is most stable in LDAO. Therefore, this construct was deemed suitable for use in biochemical assays. The apparent preference of Oep80 for LDAO is interesting given that this detergent is relatively stringent. Crystal structures of BamA in LDAO showed the protein in the inwards-open conformation where the periplasmic face of the barrel is widest, and the C-terminal β-strand is tucked into the barrel. Conversely, cryo-EM structures of BamA in DDM showed the barrel in the inwards-closed conformation where the extracellular ends of the N-terminal β-strands point away from the barrel and the periplasmic face of the barrel is constricted[42,43]. As a homologue of BamA, the conformation of Oep80 may be affected similarly by detergent identity; it could be that the inwards-closed conformation of Oep80 is less stable than the inwards-open conformation.

Our native MS analysis revealed that Oep80$^\beta$ undergoes dimerisation. Through a series of control experiments, we demonstrated that dimer formation is an inherent property of the Oep80 barrel, as opposed to an artefact related to the purification tags used. We further used crosslinking over the lateral gate or covalent attachment of a predicted β-signal peptide to occlude the $\beta_1$ strand and demonstrate that the dimer is unlikely to be a hybrid barrel. We also show there is an equilibrium between dimeric and monomeric forms, as we observed that untagged barrels can replace tagged barrels in interaction assays. The biological relevance of this monomer-dimer equilibrium will need to be investigated in future studies. Nonetheless, previous studies show some other Omp85 family members dimerise. For example, Sam50 uses a β-barrel switching mechanism to fold substrates.

When not engaged in assembly, Sam50 exists as a dimer. However, the β-signal of a substrate binds to the $\beta_1$ strand of one of the two subunits and begins to fold, and eventually this nascent barrel displaces the second Sam50[44]. If Oep80 operates in a similar manner, this would explain why the β-signal peptides we identified are able to interact with the barrel in dimeric form. Similarly, the bacterial Omp85 family member TamA of the TAM complex[45] has been shown to exist in equilibrium between monomeric and dimeric states by native PAGE, and is found as a dimer of dimers when bound to its partner TamB[46]. Some evolutionarily more distant BamA homologues, e.g., from *D. radiodurans*[47], have also shown dimerisation in structural studies. Additionally, native PAGE analysis of solubilised Arabidopsis OEMs using anti-Oep80 antibodies revealed that the protein is mostly present in a complex of around 200 kDa[18]. This size would allow for a dimer and additional accessory proteins. Loss of Crumpled Leaf (CRL) protein reduces the size to 160 kDa but this could still contain an Oep80 dimer[21]. Collectively, these observations strongly support our view that the dimerisation of Oep80 observed by native MS reflects its native state.

Robust interaction between Oep80$^\beta$ and the presumed β-signal sequences of the OEM β-barrel proteins Oep21a and Oep37 in native MS assays strongly supported the hypothesis that Oep80$^\beta$ has a role in β-barrel assembly. Importantly, mutation of the β-signals at their antepenultimate hydrophobic and penultimate residues acidic residues reduced the interactions. This is consistent with a previous study in which mutation of the C-terminal −2 and −3 residues of *Ps*Oep37 resulted in reduced formation of an Oep80/Oep37-containing complex in the OEM, and increased aggregation of Oep37 in the IMS[18]. However, direct evidence of binding between the C-terminal β-strand of a presumed substrate and Oep80, with dependency on specific residues, was previously lacking and is provided by our data. The comparatively weaker β-barrel interactions shown by representative peptides from the other three groups (i.e., TGD4, Toc75-3 and Toc159) does not necessarily indicate that these are not substrates of Oep80. They may require a chaperone or accessory protein to be brought to Oep80, or the presence of the POTRA domains to support their recognition. Alternatively, it is possible that these proteins have a β-signal that is different from those selected in our analysis of the AlphaFold structures. Moreover, our analysis focused on canonical C-terminal β-strands, but some chloroplast β-barrel proteins may use non-canonical or internal β-signals for recognition. This is analogous to the mechanism recently described in bacteria, where internal β-strands promote binding to BAM accessory proteins such as BamD[24]. Similar principles may apply in chloroplasts, though the identity of any chloroplast accessory factors remains to be elucidated. Nevertheless, the fact that the predicted β-signal peptides interacted to different extents does indicate that Oep80$^\beta$ is able to show sequence selectivity. This indicates that there is likely a consensus β-signal sequence that Oep80 uses to recognise its substrates, in the same manner as BamA and Sam50. If the first recognition step of β-barrel assembly is the same for all three Omp85 family members, it would be reasonable to assume that the downstream steps in each pathway are similar too, although this will require further investigation.

Here, we also provide experimental evidence that these β-signal peptides bind anti-parallel to the Oep80 $\beta_1$ strand, as is predicted by AlphaFold, using Oep21a $\beta_{-1}$ as a model. Of course, it is possible that this orientation is unique to Oep21a and that other peptides would not have shown the same crosslinking pattern. However, it seems likely the binding site and orientation is shared at least by other Group 1 and 2 OEPs, given their common consensus sequence. Additionally, this binding location would match that of β-signal binding to BamA and Sam50 in bacteria and mitochondria[48]. In these systems, cryo-EM and crystal structures provide conclusive evidence that β-signals hybridise in an anti-parallel orientation with the Omp85's proteins $\beta_1$ strand[26,42].

In summary, this work has used in vitro, biophysical approaches to explore the role of Oep80 as the presumed core of the chloroplastic β-barrel assembly machinery. We have presented evidence that Oep80 recognises its substrates via a consensus C-terminal β-signal sequence, revealing a marked parallel with the BAM and SAM systems. In the future, a method to express the POTRA domain of Oep80 in soluble form or to refold it without aggregation from inclusion bodies, will be highly beneficial. This will enable biochemical analysis of full-length Oep80 and a more complete assessment of the protein's function. Similarly, identification of binding partners of Oep80 will offer further insights, for example, via the reconstitution of the full complex in liposomes to probe its mechanism in detail. Previous in vivo work[18] and our in vitro characterisation here collectively highlight the importance of conserved residues of putative β-signal sequences in Oep37, Oep21 and Oep24 in recognition by Oep80 and membrane insertion. Additionally, future work must demonstrate that Oep80 binds directly to β-signals in vivo as well. Uncovering the mechanism of chloroplastic β-barrel assembly is an important prerequisite for the genetic manipulation of chloroplast β-barrel proteins, helping to ensure that attempts to improve chloroplast function and productivity do not have unintended consequences for β-barrel assembly. It also will help generate a comprehensive picture of β-barrel assembly across all domains of life, which will have far reaching implications, including in the design of novel antibiotics and understanding mitochondrial disease.

## Methods
### Constructs design
Codon-optimised sequences of AtOep80 corresponding to the full-length protein without its disordered N-terminus (residues 151–732), the β barrel domain (398–732), or the POTRA domain (156–395), were synthesised (IDT, Supplementary Table 7). These were inserted into NdeI and XhoI digested Pet28a to add an N-terminal 6×His or modified Pet15b digested with NheI and NdeI for C-terminal 6×His or GFP-6×His using In-Fusion (TaKaRa). Oep80$^\beta$ C587S/S408C/G727C and C587S/S408C were similarly synthesised and cloned into Pet28a, whereas the Oep21a $\beta_{-1}$-fused mutant was cloned into Pet15b. For the OprM signal sequence containing constructs, the OprM, His and TEV site were also synthesised and inserted with the AtOep80 sequences into Pet28a digested with XhoI and NcoI.

### Protein expression and inclusion body unfolding
Plasmids were transformed into BL21(DE3) *E. coli*. One litre LB flasks were inoculated with starter cultures, grown at 220 rpm and 37 °C until $OD_{600} = 0.6$, then induced with 0.5 M IPTG for a further 3 h at 37 °C. Overnight expression at 18 °C was also tested but all protein still formed inclusion bodies. Cultures were then pelleted at 5000 *g* for 10 min, resuspended in 20 mM Tris pH 8 and 150 mM NaCl (buffer A) and frozen. Cell pellets were resuspended with C0mplete mini protease inhibitor (Merck), sonicated and then centrifuged at 25,000 g for 30 min. Inclusion bodies were washed 3 times by resuspension in buffer A plus 0.1% Triton X-100 and re-pelleting at 25,000 g for 20 min. They were unfolded in buffer A plus 8 M urea, then centrifuged at 25,000 g for 20 min and the pellet was discarded. Urea unfolded protein was batch-bound to Ni-NTA agarose beads (Qiagen) before washing on a gravity column using buffer A plus 8 M urea and 40 mM imidazole. For proteins containing 2 or more cysteine residues, concentrated urea samples were pre-incubated with 5 mM DTT and then diluted 10 times in urea prior to purification. Elution into 2 mL fractions with buffer B and 300 mM imidazole was used to concentrate the unfolded sample.

### Protein refolding from urea
Protein-containing 8 M urea fractions were incubated with 10 mM DTT prior to refolding to reduce non-specific disulfides. Proteins containing the β-barrel domain were refolded dropwise into vigorously stirring buffer to achieve a final 8-fold dilution. For refolding conditions screens, presence or absence of 200 mM L-arginine, 150 mM or 300 mM NaCl, 1% DDM or 0.5% LDAO were all tested. Final dilution refolding buffers for Oep80$^\beta$ used 0.5% LDAO, 200 mM L-Arginine, 300 mM NaCl, 20 mM TRIS pH 8 and 2 mM DTT; using this buffer, urea sample concentrations up to 9 mg/ml could be refolded. Oep80$^{POTRA}$ was refolded by overnight dialysis using 3.5 kD cut-off dialysis tubing against 20 mM Tris, 200 mM L-Arginine, 300 mM NaCl and 2 mM DTT. Dropwise refolding was also tested.

## Concentration and purification of refolded protein

After refolding, proteins were concentrated by ultrafiltration or by ion exchange chromatography (for Oep80$^\beta$ constructs). For the latter, refolded protein was dialysed overnight to reduce NaCl concentration to 40 mM with 20 mM Tris pH 7. This was loaded onto a HiTrap Q FF cation exchange column pre-equilibrated in 20 mM Tris pH 7, 40 mM NaCl, 2 mM DTT and 0.05% LDAO. The protein was washed and then eluted in a sharp peak using 1 M NaCl and 20 mM Tris pH 8, 1 mM DTT and 0.05% LDAO. Aliquots of 500 μL were taken for SEC. For the POTRA domain, buffer A + 2 mM DTT was used, and for the β-barrel containing constructs 0.05% LDAO was also added unless other detergents are specified. SEC was conducted on a Superdex 200 Increase 10/300 GL column using AKTA-Pure™. Peak fractions were analysed by SDS-PAGE and flash-frozen before storage at −80 °C.

To generate tag-free Oep80$^\beta$, 500 μL of 6×His-Oep80$^\beta$ post cation exchange was incubated overnight with thrombin protease in a 1:25 total mass ratio. Thrombin and untagged Oep80$^\beta$ were separated using size exclusion on a Superdex 200 Increase 10/300 GL column in buffer A + 1 mM DTT and 0.05% LDAO. Peak fractions were flash frozen before storage at −80 °C.

## CD spectroscopy

Purified Oep80$^\beta$-6×His was buffer exchanged into 20 mM NaF plus either 0.05% LDAO or 0.5% C8E4 using a Biospin column. The sample was then diluted to 0.2 mg/ml. CD was conducted on a Jasco J-815 Spectropolarimeter, using the buffer alone as a blank. A 1 mm pathlength quartz cuvette was used with 1 s digital integration time. 10 accumulations over 250–190 nm were averaged. HT voltage signal was also collected in parallel. For thermal denaturation analysis, 200–250 nm scans were collected over 20–90 °C with 2 °C intervals.

## Native mass spectrometry

Before MS analysis, the proteins were buffer exchanged into 200 mM ammonium acetate pH 8.0 and 0.5% C8E4 using a Biospin-6 (BioRad) column. For the oxidised Oep80$^\beta$ C587S/S408C/G727C mutant, 0.1 mM CuSO$_4$ was added to one aliquot and pre-incubated for 4 h prior to buffer exchange. All peptides used were purchased from Proteogenix at 99% purity in powdered form. They were first dissolved in DMSO to 5 mM concentration, then diluted 10-fold into 200 mM NH$_4$CH$_3$COO plus 0.5% C8E4.

To analyse the binding of all wild type OEP β-signal peptides and the penultimate/antepenultimate residue mutants, the peptides were combined with Oep80$^\beta$-6×His to a final peptide concentration of 100 μM and protein concentration of 7 μM. To test for interaction of the cysteine mutants of Oep21a β$_{-1}$, 5 μM peptide was sprayed with approximately 3 μM protein.

Samples were introduced directly into the mass spectrometer using gold-coated capillary needles (prepared in-house). Data were collected on a Q-Exactive UHMR mass spectrometer (Thermo Fisher Scientific). The instrument parameters were as follows: capillary voltage 1.1 kV, S-lens RF 100%, quadrupole selection from 1000 to 20,000 m/z range, collisional activation in the HCD cell 100–200 V, trapping gas pressure setting 7.5, temperature 200 °C, and resolution of the instrument 12,500. The noise level was set at 3 rather than the default value of 4.64. No in-source dissociation was applied. Data were analysed using Xcalibur 4.2 (Thermo Scientific) and UniDec software packages. All experiments were repeated with similar outcomes. Statistical analysis was performed using OriginPro. Comparisons for two groups were calculated using unpaired two-tailed Student's t-tests. Exact n values are indicated in figure legends.

## Co-affinity purification

A sample of untagged Oep80$^\beta$ was combined with tagged 6×His-Oep80$^\beta$ in an approximately 1:1 molar ratio, and another was combined with buffer A + 0.05% LDAO as a control. The samples were diluted to 4 mL in the same buffer and then incubated for 4 h before loading onto separate Ni-NTA columns. Flow-through was collected and then the column was washed with 6 column volumes of 40 mM imidazole + buffer B, and then 300 mM imidazole + buffer B was used for elution. Samples were analysed by SDS-PAGE.

## Disulfide crosslinking assay

To encourage disulfide bond formation, protein aliquots post SEC purification at approximately 3 μM were incubated with 0.1 mM CuSO$_4$ at 4 °C. For peptide crosslinking, peptide was added to a 10 μM final concentration. After specified time periods, samples were boiled with 4x Laemmli loading dye with or without 50 mM DTT prior to SDS-PAGE analysis.

## Conservation analysis

ConSurf was used for conservation analysis[37,49]. The AlphaFold database structure of each OEP of interest from *A. thaliana* was used as the input. For this analysis, HMMER was used to search for homologues from UniRef90 with an *E*-value of 0.0001. The maximal % identity between sequences was 95%, and minimum was 35%. MAFFT was used to build MSAs. A sample of 150 sequences were selected for conservation score calculation using the Bayesian method.

## Statistics and reproducibility

Traces for size exclusion chromatography are representative of at least 2 repeats of the same sample and buffer conditions. CD data was averaged over 10 accumulations and is representative of 2 repeats which showed similar spectra. Native MS spectra are representative of 4 repeats. For statistical analysis of native MS results, comparisons for two groups were calculated using unpaired two-tailed Student's t-tests, with n = 4. Threshold p values are indicated in figure legends. Means, standard deviations, effect sizes, confidence intervals, exact p values and t statistics are available via Figshare https://doi.org/10.6084/m9.figshare.29733101, under Source Data for Charts and Graphs (Main Fig. 5 and Supplementary Fig. 15).

## Reporting summary

Further information on research design is available in the Nature Portfolio Reporting Summary linked to this article.

## Data availability

The authors declare that data displayed in all graphs and charts in the main manuscript and Supplementary Information as well as all native MS. raw files are available via Figshare under https://doi.org/10.6084/m9.figshare.29733101. Uncropped unedited gel images are available as Supplementary Figs. 17–20 in the Supplementary Information file. All other data are available from the corresponding author on reasonable request.

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

## Acknowledgements

We thank D. Staunton from the Molecular Biophysics Suite of the Oxford Biochemistry department for assistance with CD, as well as Aron Czako and Dušan Živković for scientific support and discussions. This research was supported by the Royal Society (URF\R1\211567 awarded to J.R.B.); UK Research and Innovation (UKRI) Frontier Research Guarantee Grant (EP/Y036158/1 awarded to J.R.B.); and UKRI Biotechnology and Biological Sciences Research Council (BB/V007300/1, BB/W015021/1, BB/X000192/1 and APP23489 awarded to R.P.J.). For the purpose of Open Access, the

## Author contributions

Conceptualisation of project: R.J.D. and J.R.B.; Methodology: R.J.D. and J.R.B.; Investigation: R.J.D. and J.R.B.; Funding acquisition: R.J.D., R.P.J., and J.R.B.; Project administration: J.R.B.; Supervision: J.R.B.; Writing – original draft: R.J.D. and J.R.B.; Writing – review and editing: R.J.D., R.P.J., and J.R.B. All authors commented on the final version of the manuscript.

## Competing interests

The authors declare no competing interests.
