## [Transparent Peer Review file · Communications Biology]

Biochemical characterisation of Oep80 supports its identity as the core component of the chloroplastic β -barrel assembly machinery

Corresponding Author: Dr Jani Bolla

Version 0:

Reviewer comments:

Reviewer #1

(Remarks to the Author)

Review Comments

This manuscript by Durant et al. analyzes β -barrel outer membrane proteins in plant chloroplasts, an area where comparative studies have lagged. It reports progress in the physicochemical analysis using recombinant proteins, for which there have been few examples to date. Furthermore, it strives to predict chloroplast-specific β -signals using structural modeling. While this paper itself may not represent a major breakthrough in the field, the method for preparing recombinant proteins achieved in this study will significantly contribute to future structural and biochemical analyses in the field of β -barrel membrane protein assembly in chloroplasts.

Major Points

1. The authors claim successful purification and refolding of the β -barrel domain of Oep80, asserting the successful formation of the Oep80 β -barrel structure. The main evidence provided is analysis using CD spectroscopy and Native MS. Firstly, while CD spectroscopy can indicate the presence of β -sheet as the major secondary structure, it cannot definitively demonstrate whether a simple β -sheet or a β -barrel has formed. Although recent reports suggest that prediction is possible using algorithms such as BeStSel, this remains uncertain. Could the authors provide more in-depth evidence that a β -barrel has been properly formed? For example, it might be possible to use classical methods for β -barrel identification such as heat modifiability, or by introducing cysteine residues at $\beta 1$ and $\beta -1$ to check for disulfide bond formation.
2. Related to point 1, the authors claim that Oep80 forms a dimer based on Native MS, as shown in Figure 3c. However, this analysis alone does not provide definitive evidence for the existence of two β -barrels formed by a single polypeptide, as proposed by the authors. For instance, how is the possibility of forming a hybrid-barrel excluded? Similar to point 1, using cysteine mutants, the presence or absence of intra-disulfide bond formation in the dimer fraction from SEC could demonstrate the association of two β -barrels.
3. The authors report that drop-wise dilution is effective for Oep80 refolding. To positively impact future research, could they show the extent of structural formation at each step? Specifically, it would be helpful to monitor and demonstrate β -barrel formation through heat modifiability or disulfide bond formation at each step.
4. Regarding the bacterial β -signals mentioned in line 68, a recent paper (DOI: 10.7554/eLife.90274.3) also discusses this, and the minimum elements required for recognition by the BAM complex have been determined. It would be beneficial to refer to this as well.

Minor Points

The manuscript uses abbreviations without explanation. For example, LADO and DDM on line 145 are not defined.

In line 387, OD600 would be preferable with OD₆₀₀ as a subscript.

In line 391, the g in 25,000 g should be italicized, and other part as well.

Reviewer #2

(Remarks to the Author)

The manuscript demonstrates that the refolded chloroplast Oep80 β -barrel domain predominantly exists as a dimer and can bind the predicted β -signal of chloroplast outer envelope proteins (OEPs) in vitro, supporting a central role for Oep80 in chloroplast β -barrel biogenesis.

The manuscript presents high-quality data and is well organized. However, the native mass spectrometry does not provide information on how Oep80 interacts with the synthetic β -signal peptides. The only supporting evidence comes from AlphaFold 3 predictions, which suggest that the first β -strand of the Oep80 β -barrel domain binds the β -signal peptides in an anti-parallel manner, similar to the interaction modes observed in BamA and Sam50. To strengthen the manuscript and support this proposed mechanism, direct experimental evidence is needed.

Reviewer #3

(Remarks to the Author)

Outer membranes of Gram-negative bacteria and endosymbiont-derived organelles like plastids and mitochondria contain several beta-barrel proteins. While it is well studied that BamA and Sam50 inserts such proteins into the outer bacterial or mitochondrial membrane, respectively, the biogenesis of plastid beta-barrel proteins is poorly understood. Initial studies revealed a role of OEP80 in this process. However, the molecular mechanisms are poorly described.

The study by Durant and colleagues addressed this question by analysing purified OEP80 and its binding to beta-signals. The authors found that overexpressed OEP80 forms a dimer and binds to beta-signal of various beta-barrel proteins with varying affinity. The presented data are solid and provide some new insights into the function of OEP80.

Minor comments:

The differential binding of OEP80 to beta-signals is interesting and could be elaborated. How many beta-barrel proteins exists in chloroplasts and are they all imported via OEP80?

Do the beta-signals also bind to OEP80 in chloroplasts? All presented studies were done with recombinantly expressed proteins.

Version 1:

Reviewer comments:

Reviewer #1

(Remarks to the Author)

The authors have addressed all of my comments thoroughly and respectfully. I have no further comments at this time. I look forward to seeing this paper published.

Reviewer #2

(Remarks to the Author)

Durant et al. have adequately addressed the suggestion by providing appropriate experimental evidence. I strongly support the publication of this work.

Reviewer #3

(Remarks to the Author)

The authors addressed my concerns in the revised version. I recommend publication of this study.

Response to Reviewers

Firstly, we would like to thank all referees for their positive remarks and constructive feedback, which helped to improve the scientific quality of the manuscript. We agree with the comments raised by all referees and have revised all relevant sections of our manuscript as outlined below.

Reviewer #1 (Remarks to the Author):

Review Comments

This manuscript by Durant et al. analyzes β -barrel outer membrane proteins in plant chloroplasts, an area where comparative studies have lagged. It reports progress in the physicochemical analysis using recombinant proteins, for which there have been few examples to date. Furthermore, it strives to predict chloroplast-specific β -signals using structural modeling. While this paper itself may not represent a major breakthrough in the field, the method for preparing recombinant proteins achieved in this study will significantly contribute to future structural and biochemical analyses in the field of β -barrel membrane protein assembly in chloroplasts.

We thank the reviewer for their feedback and thoughtful comments.

Major Points

1. The authors claim successful purification and refolding of the β -barrel domain of Oep80, asserting the successful formation of the Oep80 β -barrel structure. The main evidence provided is analysis using CD spectroscopy and Native MS. Firstly, while CD spectroscopy can indicate the presence of β -sheet as the major secondary structure, it cannot definitively demonstrate whether a simple β -sheet or a β -barrel has formed. Although recent reports suggest that prediction is possible using algorithms such as BeStSel, this remains uncertain. Could the authors provide more in-depth evidence that a β -barrel has been properly formed? For example, it might be possible to use classical methods for β -barrel identification such as heat modifiability, or by introducing cysteine residues at β 1 and β -1 to check for disulfide bond formation.

We thank the reviewer for this insightful comment and for suggesting alternative ways to support β -barrel formation.

To address this, we first assessed whether the heat modifiability method could be applied to refolded Oep80^B. Although we tested several conditions, including the absence of SDS in the loading buffer as previously shown to facilitate a band shift for TamA (Noinaj *et al.* *Methods Mol Biol*, 2015, 1328:51-56), we did not observe a clear band shift for refolded Oep80^B. As a positive control, we included the bacterial β -barrel protein, OmpT, which showed the expected heat-modifiable behaviour under identical conditions, confirming that our assay was functioning correctly (Response Figure 1).

Intrigued by this result, we further extended this assay to another chloroplast β -barrel protein, Oep21, the structure of which has been solved experimentally by NMR (Günsel *et al.* *Nat Struc Mol Biol* 2023, 30:761-769) and is known to adopt a β -barrel fold. We replicated the published refolding method, and tested heat modifiability of refolded Oep21 on the same semi-native gels. Surprisingly, Oep21 also did not display heat modifiability in our assays (Response Figure 1), suggesting that this property may not be conserved across all β -barrel proteins, especially those of organellar origin. In line with this, previous reports have shown that some mitochondrial β -barrel proteins, e.g., VDAC do not show heat modifiability either (Shi *et al.* *Biochem Biophys Res Commun* 2003, 303:475-82). The differing properties of organellar

proteins in this regard may reflect the unique chemical compositions of organellar membranes and corresponding differences in the physicochemical properties of membrane-embedded organellar proteins (Henderson et al. *Annu. Rev. Microbiology* 2016, 70:255-278; Inoue, *JIPB* 2007, 49:1100-1111)

Response Figure 1. Oep80^β and Oep21 do not show heat modifiability, unlike bacterial β -barrel OmpT. **a)** Semi-native SDS-PAGE gel analysis of refolded proteins, stained with Coomassie blue. Streaking in Oep21a lanes was due to high ionic strength from residual guanidinium hydrochloride. **b)** Anti-His western blot of the same gel. U and F indicate unfolded and folded OmpT, respectively; D indicates dimer bands. 1% SDS refers to the presence (+) or absence (-) of SDS in sample loading buffer.

As an alternative approach (as suggested by the reviewer), we generated a mutant of Oep80^β that has native cysteine C587 removed and two new cysteines introduced in the β_1 and β_{-1} strands (C587S/S408C/G727C). Mutation of BamA residues G433 and N805 to cysteine had previously been used to form a gate-bridging disulfide (Noinaj *et al.* *Structure* 2014, 22:1055-62). Comparison of the Oep80 AlphaFold model with the established BamA structure suggested S408 and G727, respectively, as the corresponding residues in Oep80, and so these were replaced with cysteine. Mutation of C587 ensured any intra-strand disulfide observed was between the newly introduced cysteines. Strikingly, AlphaFold 3 spontaneously predicted a gate-bridging disulfide bond for this mutant (Response Figure 2a).

Response Figure 2. Oep80^β C587S/S408C/G727C forms an intra-strand disulfide when oxidised, which may span the lateral gate. a) AlphaFold 3 model of this mutant, showing a lateral gate-spanning disulfide bond in yellow. Introduced β_1 and β_{-1} cysteines are coloured in red. pTM = 0.84. b) SDS-PAGE of Oep80^β C587S/S408C/G727C following SEC purification (No Cu²⁺) and after incubation at 4°C with 0.1 mM CuSO₄ overnight (+ Cu²⁺). +DTT indicates that the sample was boiled with 50 mM DTT in loading dye. c) SDS-PAGE of Oep80^β C587S/S408C/G727C in unfolded form (U) and at timepoints during refolding. The normal refolding protocol was used, but with the final DTT concentration lowered to 0.5 mM and the addition of 100 μ M CuSO₄ to the refolding buffer.

This C587S/S408C/G727C mutant was expressed, refolded, and purified in the same manner as wild-type Oep80^β. We then monitored disulphide bond formation by SDS-PAGE under reducing and non-reducing conditions (protocol adapted from Kuszak *et al.* Methods Mol Biol 2015, 1329:137-47). A faster-migrating band was observed in the absence of DTT, consistent with the formation of an intramolecular disulphide bond between the added cysteines (Response Figure 2b). This band disappeared upon DTT treatment, confirming that the cysteines are in close proximity in the folded state and able to form a disulphide bond (Response Figure 2b). Taken together, while heat modifiability was not observed for Oep80 or Oep21, our cysteine crosslinking data provide strong evidence that a β -barrel has been formed properly.

This data is now included in Figure 3, and the main text is modified as follows:

“To confirm the barrel closure and to determine whether the observed dimer forms a hybrid barrel or two independent β barrels, we generated a mutant of Oep80^β that has native cysteine C587 removed and 2 new cysteines introduced in the β_1 and β_{-1} strands (C587S/S408C/G727C) (Fig 3d). Mutation of BamA residues G433 and N805 to cysteine had previously been shown to form a gate-bridging disulfide³³. Comparison of Oep80’s AlphaFold model with BamA’s known structure suggested S408 and G727, respectively, as the corresponding residues in Oep80. Strikingly, AlphaFold 3 spontaneously predicted a gate-bridging disulfide bond for the C587S/S408C/G727C mutant (Fig 3e). After refolding and purification of this mutant, we then monitored disulphide bond formation on SDS-PAGE under reducing and non-reducing conditions³⁴. A faster-migrating band was observed in the absence of DTT, consistent with the formation of an intramolecular disulphide bond between the added cysteines (Fig 3f). This band disappeared upon DTT treatment. This confirmed that the cysteines are in close proximity in the folded state and form intra-strand disulfide bonds, which likely span the lateral gate.”

2. Related to point 1, the authors claim that Oep80 forms a dimer based on Native MS, as shown in Figure 3c. However, this analysis alone does not provide definitive evidence for the existence of two β -barrels formed by a single polypeptide, as proposed by the authors. For instance, how is the possibility of forming a hybrid-barrel excluded? Similar to point 1, using cysteine mutants, the presence or absence of intra-disulfide bond formation in the dimer fraction from SEC could demonstrate the association of two β -barrels.

We thank the reviewer for raising this important point. To investigate this, we leveraged the Oep80 ^{β} C587S/S408C/G727C mutant (β_1 & β_{-1} cysteine) described above in point 1. In a hybrid barrel configuration, crosslinking across the gate would block β_1 strand availability and thus be expected to interfere with inter-subunit strand exchange. If the observed dimer resulted from a hybrid barrel, we would expect Cu²⁺-mediated disulphide formation to disrupt this interaction and shift the equilibrium toward the monomeric species (Response Figure 3a). However, native MS analysis of Oep80 ^{β} C587S/S408C/G727C both before (Response Figure 3b) and after (Response Figure 3c) Cu²⁺ incubation showed no significant presence of a monomeric form of the protein. This supports the hypothesis that the observed dimer is not a large hybrid barrel but instead two independent barrels that associate together in solution.

To further validate this conclusion, we generated an Oep80 ^{β} mutant with the Oep21a β_{-1} peptide sequence covalently attached via a flexible GS linker. The single native cysteine residue of Oep80 ^{β} was also mutated (to serine). In this construct, the covalently attached β_{-1} peptide is expected to occupy the β_1 binding groove, mimicking the interaction observed in our previous peptide-binding native MS experiments. If the Oep80 dimer were a hybrid barrel formed by inter-subunit strand exchange, this steric occlusion should reduce dimerisation. Nevertheless, native MS revealed that this fusion protein also exists predominantly as a dimer, with undetectable monomer signal above baseline noise (Response Figures 3d, 3e). This reinforces the conclusion that the Oep80 dimer comprises two independently folded β -barrels that associate non-covalently, rather than a single hybrid structure formed by strand-swapping.

Response Figure 3. Occluding the Oep80 β_1 strand via crosslinking or covalent peptide addition does not increase monomer abundance, suggesting the observed Oep80 β dimer is not a hybrid barrel. **a)** Schematic to show how a cross-gate disulphide bond should prevent hybrid barrel formation. Red circles represent introduced cysteine residues and yellow represents disulfide crosslinking. **b,c)** Representative native mass spectra of Oep80 β C587S/S408C/G727C after purification (**b**), and after purification and overnight 4°C incubation with 0.1 mM CuSO₄ (**c**). **d)** Schematic to show how an N-terminally linked Oep21a β_1 peptide (green) might disrupt hybrid barrel formation. **e)** Representative native mass spectrum of Oep21a β_1 -fused-Oep80 β . See Table S4 for predicted and observed masses.

These results are now discussed in the main text as follows:

“Oep80 β dimer is not a hybrid-barrel

...
In a hybrid-barrel configuration, crosslinking across the gate would block β_1 strand availability and thus be expected to interfere with inter-subunit strand exchange. If the observed dimer resulted from a hybrid barrel, we would expect Cu²⁺-mediated disulphide formation to disrupt this interaction and shift the equilibrium toward the monomeric species (Fig 3d). However, native MS analysis of Oep80 β C587S/S408C/G727C both before (Fig S8a) and after (Fig S8b)

Cu²⁺ incubation showed no significant presence of a monomeric form of the protein. This suggests that the observed dimer is not a large hybrid-barrel but instead two independent barrels that are associated in solution.

And:

“Having found that Oep21a β –1 interacts strongly with Oep80 β , we used this peptide to further validate our conclusion that the Oep80 β dimer does not form a hybrid barrel. We generated an Oep80 β mutant with the Oep21a β –1 peptide sequence covalently attached via a flexible GS linker. This mutant also had the Oep80 β 's one native cysteine residue mutated to serine. In this construct, the covalently attached β -1 peptide is expected to form an anti-parallel β -sheet with the β 1 strand, mimicking the interaction observed in our peptide-binding native MS experiments. If the Oep80 dimer were a hybrid barrel formed by inter-subunit strand exchange, this steric occlusion should reduce dimerisation. Native MS revealed that this fusion protein also exists as a dimer (Fig S14). This reinforces the conclusion that the Oep80 dimer comprises two independently folded β -barrels that associate non-covalently, rather than a single hybrid structure formed by strand-swapping.”

In Discussion:

“Our native MS analysis revealed that Oep80 β undergoes dimerisation. Through a series of control experiments, we demonstrated that dimer formation is an inherent property of the Oep80 barrel, as opposed to an artefact related to the purification tags used. We further used crosslinking over the lateral gate or covalent attachment of a predicted β -signal peptide to occlude the β _i strand and demonstrate that the dimer is unlikely to be a hybrid barrel. We also show there is an equilibrium between dimeric and monomeric forms, as we observed that untagged barrels can replace tagged barrels in interaction assays.”

3. The authors report that drop-wise dilution is effective for Oep80 refolding. To positively impact future research, could they show the extent of structural formation at each step? Specifically, it would be helpful to monitor and demonstrate β -barrel formation through heat modifiability or disulfide bond formation at each step.

We thank the reviewer for this suggestion. Ideally, refolding would be most easily monitored by heat modifiability; however, as shown in Response Figure 1, Oep80 does not display this behaviour, limiting the utility of this method for time-resolved analysis. Monitoring a time course of refolding via disulfide bond formation (using the cysteine mutant) is somewhat difficult. This is because effective refolding requires reducing conditions to prevent premature intra-strand disulphide formation in the unfolded state. As seen in Response Figure 2c, the unfolded protein aggregates heavily when oxidised prematurely, highlighting the importance of maintaining a reducing environment during initial stages of folding.

To examine whether disulphide formation might nonetheless provide insight into the refolding timeline, we added 0.1 mM CuSO₄ to the refolding buffer under mildly reducing conditions (1 mM DTT) and took samples at defined timepoints. As shown in Response Figure 2c, a faster-migrating band corresponding to the disulfide-crosslinked form appears by 10 minutes and increases gradually over time. While the exact band intensity reflects both the folding status and the redox potential of the buffer, this assay suggests that a substantial portion of the protein folds into a conformation compatible with β -barrel closure within the first 10-30 minutes. Despite this, we typically allow the refolding process to proceed overnight under gentle agitation to ensure completion, especially for downstream purification and functional analysis.

We added the following to the revised version:

“After refolding and purification of this mutant, we then monitored disulphide bond formation on SDS-PAGE under reducing and non-reducing conditions³⁴. A faster-migrating band was observed in the absence of DTT, consistent with the formation of an intramolecular disulphide bond between the added cysteines (Fig 3f). This band disappeared upon DTT treatment. This confirmed that the cysteines are in close proximity in the folded state and form intra-strand disulfide bonds, which likely span the lateral gate. We also attempted to track folding kinetics using disulphide crosslinking but found that the need for reducing conditions during refolding complicated time-resolved quantification. Nevertheless, crosslinking becomes detectable within 10 minutes, suggesting rapid folding similar to that observed with dilution refolding of bacterial barrel proteins³⁵.”

4. Regarding the bacterial β -signals mentioned in line 68, a recent paper (DOI: 10.7554/eLife.90274.3) also discusses this, and the minimum elements required for recognition by the BAM complex have been determined. It would be beneficial to refer to this as well.

We thank the reviewer for this suggestion. We now cite this work both in the Introduction and Discussion as follows:

“Omp85 barrels have a dynamic lateral gate, which enables recognition and binding of β -signal sequences to their first β -strand (β_1) (Fig 1b)^{6,22}. These sequences are found in the C-terminal β -strand of substrate barrels (β_{-1}). Sam50 recognises the consensus sequence PoXGXX φ X φ , while BAM recognises X φ X φ XYXF (φ indicates hydrophobic, Po indicates polar)²³. A recent study further demonstrated that internal sequences, termed internal β -signals, also contribute to substrate recognition by the BAM complex via interactions with BamD, suggesting a more complex, distributed recognition mechanism than previously envisaged²⁴.”

“The comparatively weaker β -barrel interactions shown by representative peptides from the other three groups (i.e., TGD4, Toc75-3 and Toc159) does not necessarily indicate that these are not substrates of Oep80. They may require a chaperone or accessory protein to be brought to Oep80, or the presence of the POTRA domains to support their recognition. Alternatively, it is possible that these proteins have a β -signal that is different from those selected in our analysis of the AlphaFold structures. Moreover, our analysis focused on canonical C-terminal β -strands, but some chloroplast β -barrel proteins may use non-canonical or internal β -signals for recognition. This is analogous to the mechanism recently described in bacteria, where internal β -strands promote binding to BAM accessory proteins such as BamD²⁴. Similar principles may apply in chloroplasts, though the identity of any chloroplast accessory factors remains to be elucidated.”

Minor Points

The manuscript uses abbreviations without explanation. For example, LADO and DDM on line 145 are not defined.

In line 387, OD600 would be preferable with OD₆₀₀ as a subscript.

In line 391, the g in 25,000 g should be italicized, and other part as well.

We thank the reviewer for pointing out these errors. They have been corrected in the revised manuscript.

Reviewer #2 (Remarks to the Author):

The manuscript demonstrates that the refolded chloroplast Oep80 β -barrel domain predominantly exists as a dimer and can bind the predicted β -signal of chloroplast outer envelope proteins (OEPs) *in vitro*, supporting a central role for Oep80 in chloroplast β -barrel biogenesis.

The manuscript presents high-quality data and is well organized. However, the native mass spectrometry does not provide information on how Oep80 interacts with the synthetic β -signal peptides. The only supporting evidence comes from AlphaFold 3 predictions, which suggest that the first β -strand of the Oep80 β -barrel domain binds the β -signal peptides in an anti-parallel manner, similar to the interaction modes observed in BamA and Sam50. To strengthen the manuscript and support this proposed mechanism, direct experimental evidence is needed.

We thank the reviewer for this insightful and valuable suggestion.

To directly test the location and orientation of β -signal peptide binding to Oep80 ^{β} , we generated a cysteine mutant of Oep80 ^{β} (C587S/S408C), placing the cysteine at the C-terminal end of the β_1 strand. We then synthesised two versions of the Oep21a β_1 signal peptide: one with an added C-terminal cysteine (CterC; GKWNVRYLC) and one with K160 toward the N-terminus mutated to cysteine (KtoC; GCWNVRYL). If the peptide binds to the β_1 strand in an anti-parallel orientation, as predicted by AlphaFold 3, only the KtoC peptide should be able to form a disulphide bond with the S408C site (Response Figure 4a). This prediction was further supported by AlphaFold 3 structural models of the mutant Oep80 ^{β} in complex with KtoC (Response Figure 4b), which show a plausible lateral gate interaction and a predicted disulphide bridge between the introduced cysteines.

Response Figure 4. AlphaFold 3 predicts disulphide bond formation between Oep80 ^{β} C587S/S408C and an Oep21a β -signal mutant containing an N-terminal cysteine (Oep21a β -1 KtoC). a) Schematic showing the experimental strategy. Combining Oep80 ^{β} C587S/S408C with mutant Oep21a β_1 peptides containing cysteines at either end enables differentiation between parallel and antiparallel signal peptide binding. Green arrow represents the Oep21 peptide; red circles represent introduced cysteine residues; and yellow lines represent disulfide bonds. b) AlphaFold 3 model of Oep80 ^{β} C587S/S408C plus Oep21a β_1 KtoC. The insets show the interaction interface in detail in comparison with wild-type Oep80 ^{β} plus wild-type Oep21a β_1 . Peptide backbones are in green; the Oep80 barrel backbone is in cyan. Mutant residues are coloured red. Dotted lines in the insets indicate likely hydrogen bonding interactions, with distances in Å.

Consistent with this model, incubation of Oep80 ^{β} C587S/S408C with each peptide in the presence of CuSO₄ to promote oxidation led to disulphide crosslink formation with the KtoC peptide but not with CterC (Response Figure 5a). A higher molecular weight band was observed by SDS-PAGE only with the KtoC mutant, and this band was reduced by DTT treatment confirming disulphide bond formation.

To ensure that the absence of crosslinking with CterC was not due to reduced binding affinity, we performed native MS using both peptides. We observed that both KtoC and CterC peptides shifted the main Oep80 ^{β} charge state and produced additional peptide-bound peaks of similar intensity, indicating comparable non-covalent binding (Response Figure 5c, 5d). This supports

the interpretation that both peptides bind to the same site, but only KtoC positions its cysteine close enough to form a disulphide with Oep80 β_1 -S408C.

These results provide direct experimental support for the predicted anti-parallel binding mode of β -signal peptides to Oep80 $^\beta$ and confirm that binding occurs at the β_1 strand, analogous to the mechanism used by BamA and Sam50.

Response Figure 5. Oep80 $^\beta$ C587S/S408C crosslinks efficiently with Oep21a β_1 KtoC unlike Oep21a β_1 CterC, despite both binding non-covalently. a) Oep80 $^\beta$ S408C/C587S alone or plus either peptide was incubated at 4°C with 0.1 mM CuSO₄. Samples were taken after 3 hours (3h) or after overnight (O/N) incubation. +DTT indicates that the sample was boiled with 50 mM DTT in loading dye. **b-d)** Native mass spectra showing: Oep80 $^\beta$ alone (b), Oep80 $^\beta$ + KtoC peptide (c), and Oep80 $^\beta$ + CterC peptide (d). Peptide-bound species are visible in both c and d with similar intensity, indicating similar binding efficiencies. See Table S4 and Table S6 for predicted and observed masses.

These results are now included in the revised version as figure 6 and the main text is modified as:

“ β -signal peptide appears to bind anti-parallel to Oep80’s β_1 strand

To directly test the location and orientation of β -signal peptide binding to Oep80 ^{β} , we generated a cysteine mutant of Oep80 ^{β} (C587S/S408C), placing the cysteine at the C-terminal end of the β_1 strand. We then synthesised two versions of the Oep21a β_{-1} signal peptide: one with a C-terminal cysteine (CterC; GKWNVRYLC) and one with K160 toward the N-terminus mutated to cysteine (KtoC; GCWNVRYL). If the peptide binds to the β_1 strand in an anti-parallel orientation, as predicted by AlphaFold 3, only the KtoC peptide should be able to form a disulphide bond with the S408C site (Fig 6a). This prediction was further supported by AlphaFold 3 structural models of the mutant Oep80 ^{β} in complex with KtoC (Fig 6b), which show a plausible lateral gate interaction and a predicted disulphide bridge between the introduced cysteines.

Consistent with this model, incubation of Oep80 ^{β} C587S/S408C with each peptide in the presence of CuSO₄ to promote oxidation led to disulphide crosslink formation with the KtoC peptide but not with CterC (Fig 6c). A higher molecular weight band was observed by SDS-PAGE only in the KtoC condition, and this band was reduced by DTT treatment, confirming disulphide bond formation.

To ensure the absence of crosslinking with CterC was not due to reduced binding affinity, we performed native MS analysis using both peptides. We observed the charge state series corresponding to both KtoC and CterC binding with similar intensity, indicating comparable non-covalent binding (Fig S16). This supports the interpretation that both peptides bind to the same site, but only KtoC positions its cysteine close enough to form a disulphide with Oep80 β_1 -S408C.”

And in Discussion:

Here we also provide experimental evidence that these β -signal peptides bind anti-parallel to the Oep80 β_1 strand, as is predicted by AlphaFold, using Oep21a β_{-1} as a model. Of course, it is possible that this orientation is unique to Oep21a and that other peptides would not have shown the same crosslinking pattern. However, it seems likely the binding site and orientation is shared at least by other Group 1 and 2 OEPs, given their common consensus sequence. Additionally, this binding location would match that of β -signal binding to BamA and Sam50 in bacteria and mitochondria⁴⁸. In these systems, cryo-EM and crystal structures provide conclusive evidence that β -signals hybridise in an anti-parallel orientation with the Omp85’s proteins β_1 strand^{26,42}.

Reviewer #3 (Remarks to the Author):

Outer membranes of Gram-negative bacteria and endosymbiont-derived organelles like plastids and mitochondria contain several beta-barrel proteins. While it is well studied that BamA and Sam50 inserts such proteins into the outer bacterial or mitochondrial membrane, respectively, the biogenesis of plastid beta-barrel proteins is poorly understood. Initial studies revealed a role of OEP80 in this process. However, the molecular mechanisms are poorly described.

The study by Durant and colleagues addressed this question by analysing purified OEP80 and its binding to beta-signals. The authors found that overexpressed OEP80 forms a dimer and binds to beta-signal of various beta-barrel proteins with varying affinity. The presented data are solid and provide some new insights into the function of OEP80.

We thank the reviewer for their positive remarks.

Minor comments:

The differential binding of OEP80 to beta-signals is interesting and could be elaborated. How many beta-barrel proteins exist in chloroplasts and are they all imported via OEP80?

We thank the reviewer for highlighting this important point. Currently, around 16 β -barrel proteins have been identified in the chloroplast outer envelope membrane of the model plant *Arabidopsis thaliana*. We have sorted these into four functional groups: Group 1 - solute transporters (Oep37, Oep21, Oep24 and Oep40); Group 2 - lipid transporters of the LptD family (Lptd1 and Tgd4); Group 3 - Omp85 family homologues (Toc75-III, Toc75-IV, SP2/p39 and Oep80); and finally, Group 4 - Toc159 family proteins which co-assemble with Toc75 within the TOC complex (Toc159, Toc132, Toc90 and Toc120).

Previous *in vivo* work has provided evidence that Oep80 is required for the proper insertion of at least some Group 1 and Group 3 members, including Oep37, p39/SP2 and Toc75 (Gross *et al.* Plant Cell 2021, 33:1657-1681; Huang *et al.* Plant Physiol 2011, 157(1):147-59.). While additional factors may participate in this process, there is currently no known alternative β -barrel assembly machinery in plastids. Therefore, it is likely that Oep80 is responsible for the biogenesis of the majority, if not all, of these β -barrel proteins. We have clarified and expanded on this point in the revised manuscript.

We have included a more explicit discussion of other β -barrel proteins found in chloroplasts in our revised manuscript:

On page 2

*“The core components of TOC and CHLORAD are β -barrel proteins, namely Toc75 (which co-assembles with Toc159) and SP2 (p39), respectively¹². Additionally, the chloroplast outer envelope membrane (OEM) contains several other β -barrel proteins (e.g., Oep21, Oep24, Oep37, Tgd4) required for metabolite and solute transport¹³⁻¹⁷ adding to around 16 β -barrel proteins identified in *A.thaliana*. How these β -barrel proteins reach the chloroplast and assemble in the OEM is largely unknown.”*

On page 3

“Additionally, mutating the antepenultimate hydrophobic residue and penultimate acidic residue in the β_1 strand of Oep37 reduced the amount of Oep37 present in these Oep80-containing complexes, indicating these residues may be important for substrate recognition by Oep80. However, such a signal in other β -barrel proteins has not been identified. Identification of such a signal will help clarify whether all the chloroplastic β -barrel proteins biogenesis is mediated by Oep80; currently, there is no known alternative β -barrel assembly machinery in plastids. Furthermore, more evidence is needed to definitively show that Oep80 recognises and binds β -signals in substrate barrel proteins, and further characterisation of the β -signal consensus sequence is also required.”

Do the beta-signals also bind to OEP80 in chloroplasts? All presented studies were done with recombinantly expressed proteins.

We thank the reviewer for raising this important question. Previous *in vivo* work (Gross *et al.* Plant Cell 2021, 33:1657-1681) has elegantly shown that mutations in conserved residues of putative β -signal sequences in Oep37 and Oep24 impair their membrane insertion and reduce

interaction with Oep80, suggesting these signals are functionally important. However, direct binding between the β -signals and Oep80 was not demonstrated.

Thus, our *in vitro* assays were designed to directly test β -signal binding to the Oep80 β -barrel domain and clarify the orientation and specificity of this interaction. Nonetheless, we fully agree that confirming these interactions in a native chloroplast context is a crucial next step. We have now emphasised this point more clearly in the discussion section of the revised manuscript.

*“Similarly, identification of binding partners of Oep80 will offer further insights, for example, via the reconstitution of the full complex in liposomes to probe its mechanism in detail. Previous *in vivo* work¹⁸ and our *in vitro* characterisation here collectively highlight the importance of conserved residues of putative β -signal sequences in Oep37, Oep21 and Oep24 in recognition by Oep80 and membrane insertion. Additionally, future work must demonstrate that Oep80 binds directly to β -signals *in vivo* as well.”*

REVIEWERS' COMMENTS:

Reviewer #1 (Remarks to the Author):

The authors have addressed all of my comments thoroughly and respectfully. I have no further comments at this time. I look forward to seeing this paper published.

Reviewer #2 (Remarks to the Author):

Durant et al. have adequately addressed the suggestion by providing appropriate experimental evidence. I strongly support the publication of this work.

Reviewer #3 (Remarks to the Author):

The authors addressed my concerns in the revised version. I recommend publication of this study.

We thank all three reviewers for their positive remarks and valuable contributions toward improving the manuscript for publication.